# The Interplay of Mycosporine-like Amino Acids between Phytoplankton Groups and Northern Krill (*Thysanoessa* sp.) in a High-Latitude Fjord (Kongsfjorden, Svalbard)

**DOI:** 10.3390/md20040238

**Published:** 2022-03-29

**Authors:** Bo Kyung Kim, Mi-Ok Park, Jun-Oh Min, Sung-Ho Kang, Kyung-Hoon Shin, Eun Jin Yang, Sun-Yong Ha

**Affiliations:** 1Division of Polar Ocean Sciences, Korea Polar Research Institute, Incheon 21990, Korea; bkkim@kopri.re.kr (B.K.K.); jomin@kopri.re.kr (J.-O.M.); shkang@kopri.re.kr (S.-H.K.); ejyang@kopri.re.kr (E.J.Y.); 2Department of Oceanography, Pukyong National University, Busan 48513, Korea; mopark@pknu.ac.kr; 3Department of Marine Sciences and Convergent Engineering, Hanyang University, Ansan 15588, Korea; shinkh@hanyang.ac.kr

**Keywords:** pigment, mycosporine-like amino acids, phytoplankton, krill, Kongsfjorden, Svalbard

## Abstract

We investigated pigment and mycosporine-like amino acid (MAA) concentrations of phytoplankton and Northern krill (*Thysanoessa* sp.) in sub-Arctic Kongsfjorden. Chlorophyll a (Chl-a) concentrations in the surface and middle-layer water were 0.44 μg L^−1^ (±0.17 μg L^−1^) and 0.63 μg L^−1^ (±0.25 μg L^−1^), respectively. Alloxanthin (Allo, a marker of cryptophytes) was observed at all stations, and its mean values for surface and middle-layer water were 0.09 μg L^−1^ (±0.05 μg L^−1^) and 0.05 (±0.02 μg L^−1^), respectively. The mean MAA-to-Chl-a ratios at the surface (3.31 ± 2.58 μg (μg Chl-a)^−1^) were significantly higher than those in the middle-layer water (0.88 ± 0.49 μg (μg Chl-a)^−1^), suggesting that these compounds play an important role in reducing UV photodamage. In gut pigment levels of Northern krill, the most abundant accessory pigment was Allo (2.79 ± 0.33 μg g^−1^ dry weight; d.w.), as was the accumulation of Chl-a (8.29 ± 1.13 μg g^−1^ d.w.). The average concentration of MAAs was 1.87 mg g^−1^ d.w. (±0.88 mg g^−1^ d.w.) in krill eyes, which was higher than that in all other body parts (0.99 ± 0.41 mg g^−1^ d.w.), except for the gut. *Thysanoessa* sp. was found to contain five identified MAAs (shinorine, palythine, porphyra-334, mycosporine-glycine, and M-332) in the krill eye, whereas shinorine and porphyra-334 were only observed in the krill body, not the eyes and gut. These findings suggest that Northern krill accumulate MAAs of various compositions through the diet (mainly cryptophytes) and translocate them among their body parts as an adaptation for photoprotection and physiological demands.

## 1. Introduction

The reduction in stratospheric ozone in the Arctic has led to enhanced ultraviolet radiation (UV-R), which affects marine organisms such as phytoplankton and macroalgae [1,2,3,4,5,6,7]. Many studies have revealed an inhibition of photosynthesis and primary production in phytoplankton and DNA damage, as well as cellular protein synthesis after exposure to solar UV-R [8,9,10,11,12,13,14]. In particular, cell metabolism and survival vary among species of phytoplankton under UV-R because photoadaptation and DNA damage-associated repair mechanisms are species-specific [9,15].

Generally, living organisms survive in extreme environments by expressing various light protection substances and reacting with control mechanisms after exposure to strong light or UV-R [16]. Likewise, phytoplankton can be protected or mitigated through the synthesis of UV-R-absorbing compounds and photoprotective pigments against UV-R [14,17]. In particular, mycosporine-like amino acids (MAAs) are known to be UV-R-absorbing compounds. They are small water-soluble molecules (<400 Da), and contain aminocyclohexenone or aminocyclohexene imine with absorption maxima between 310 and 360 nm [18,19,20]. MAA compounds are found in primary marine and freshwater producers such as cyanobacteria [18,21,22,23,24,25,26]. The content of MAAs can vary depending on not only level of UV exposure, but also physicochemical stressors such as nitrogen, temperature, and salinity. These UV-R-absorbing compounds have also been demonstrated in antioxidants, anti-inflammatory, and osmotic regulation [27,28]. In this regard, research on the presence of MAA as biologically active compounds in various organisms contributes to various fields such as functional food and natural sunscreen/photoprotective cosmetics for humans [29,30,31]. Recent studies have focused on MAAs as a source of natural bioactive compounds against synthetic sunscreens [27,32].

The presence of MAAs in zooplankton are mostly considered to be photoprotection function. Zooplankton are not capable of synthesizing MAAs; they can absorb MAAs from their diet (phytoplankton) [16,33]. Among them, euphausiids (krill) maintain a central trophic position in aquatic food webs because they feed on phytoplankton and provide a food source for fish, birds, and mammals [34]. Krill have evolved various strategies to cope with high fluxes of incoming solar radiation. For instance, these organisms possess defense mechanisms that act to prevent UV-Rs, such as active diel vertical migration [35], the accumulation of photoprotective compounds such as MAA and carotenoids [36,37], or primary mediation by the photoenzymatic repair system [38].

In the fjords of Spitsbergen, the krill community is dominated by *Thysanoessa* sp. (*Thysanoessa inermis*, *Thysanoessa raschii*, and *Thysanoessa longicaudata*) [39,40]. Among *euphausiids*, the key species are *T. inermis* and *T. raschii*, accounting for 71% of the *euphausiid* biomass in Kongsfjorden [33]. This study investigated the UV-absorbing material MAAs in phytoplankton and *Thysanoessa* sp. in Kongsfjorden. Here, we assumed that the *Thysanoessa* sp. (hereafter, “Northern krill”) acquired UV-R- absorbing MAAs from dietary phytoplankton during the sampling period. In addition, we analyzed phytoplankton taxonomy using photosynthetic biomarker pigments. This tool was useful to assess ecological characteristics related to the possibility of identifying the diversity of phytoplankton groups and their reaction to various environmental conditions [41,42,43,44,45]. Furthermore, gut pigment analyses can be used to identify dominant food sources and provide information on feeding selectivity in various taxa [46,47]. Therefore, our objectives in the present study were: (1) to assess the distribution of MAAs in the body of Northern krill; (2) to identify the dominant food source for Northern krill based on pigment analysis; and (3) to infer the relationship between Northern krill and their diet (phytoplankton) in the Kongsfjorden pelagic food web.

## 2. Results

### 2.1. Chlorophyll a and Accessory Pigments of Phytoplankton

Among the five pigments (including chlorophyll a), four pigments (peridinin, Per; fucoxanthin, Fuco; alloxanthin, Allo; and chlorophyll a, Chl-a) were detected in the surface and middle-layer waters of Kongsfjorden, with mean values of 0.04 μg L^−1^ (standard deviation (SD) = ±0.01 μg L^−1^), 0.38 μg L^−1^ (SD = ±0.24 μg L^−1^), 0.07 μg L^−1^ (SD = ±0.04 μg L^−1^), and 0.5 μg L^−1^ (SD = ±0.2 μg L^−1^), respectively (Figure 1). Phytoplankton biomass, reflected by the Chl-a concentration, ranged from 0.1 to 1.1 μg L^−1^ in Kongsfjorden. The highest mean values of Chl-a for the surface and middle-layer water were observed at T3 (0.9 μg L^−1^) and A4 (1.1 μg L^−1^), respectively. Overall, Chl-a concentrations for the surface layer at transect T adjacent land stations (T1, T2, T3, T4, and T5) were high compared with the middle-layer, whereas Chl-a at the remnant stations was, on average, twofold higher in the middle-layer than in the surface layer.

The transects also showed differences in accessory pigment compositions between the surface layer and the middle layer. Indeed, Allo among the accessory pigments was found in the surface and middle-layer waters throughout the fjord, whereas Fuco was detected only for several stations (A1, A2, A3, and A4). In the surface water, the Fuco concentration increased from the inner fjord to the outer fjord, whereas Allo showed a decreasing trend. (Figure 1). Allo showed higher concentrations at the surface (0.09 ± 0.05 μg L^−1^) than in the middle layer (0.05 ± 0.02 μg L^−1^), while the mean concentration of algal pigment Fuco at the surface (0.2 ± 0.1 μg L^−1^) was three times lower than that in the middle-layer (0.6 ± 0.2 μg L^−1^) waters, with a maximum value of 0.80 μg L^−1^ at A3 (30 m depth). Per was mainly distributed in the middle-layer water, and it was not observed in the surface water at transect A. As a result, the contribution of Per reached 66.3% of total accessory pigments at T3, accounting for 54.5% of total accessory pigments in the middle-layer water of adjacent land stations.

The overall compositions within 30 m depth showed that Fuco alone accounted for 50% to 96.6% of the total accessory in transect A. Conversely, in transect T, the pigments were characterized by a larger contribution of Allo (72.2% of total accessory pigments) and by a moderate proportion of Per (27.8% of total accessory pigments).

### 2.2. Mycosporine-like Amino Acid (MAA) Concentrations of Phytoplankton between Two Water Layers

The concentrations of individual MAA compounds of phytoplankton varied with depth and sampling sites. In the surface water, five distinct MAAs (shinorine; λmax = 334 nm SH, palythine; λmax = 320 nm, PA, porphyra-334; λmax = 334 nm PR, mycosporine-glycine; λmax = 310 nm, MG, and asterina-330; λmax = 330 nm, AS) were detected in natural phytoplankton, whereas four (SH, PA, PR, and MG) out of the five MAAs were isolated in the middle-layer samples (Table 1).

The surface-water phytoplankton community had significantly higher concentrations of SH (0.41 ± 0.32 μg L^−1^) and PA (0.48 ± 0.47 μg L^−1^) than those from the middle-layer water (0.06 ± 0.04 μg L^−1^ and 0.06 ± 0.03 μg L^−1^, respectively) (*t*-test, *p* < 0.05) (Table 1). AS contributed ~1% of the total MAA content in the surface phytoplankton, and it was not found in the middle layer. MG among individual MAAs was predominant in the middle-layer throughout the fjord (Figure 2).

### 2.3. Distribution of Pigments and Mycosporine-like Amino Acids (MAAs) in Northern Krill

For the distribution of accessory pigments in Northern krill, Fuco (0.016 ± 0.002 μg g^−1^ d.w.) and 19′-hexanolyoxyfucoxanthin (19′-Hex, 0.029 ± 0.009 μg g^−1^ d.w.) were present in trace amounts, whereas Allo exhibited relatively high concentrations (2.791 ± 0.333 μg g^−1^ d.w.) (Table 2). Chl-a was also present at 8.29 μg g^−1^ d.w. (±1.13 μg g^−1^ d.w.) in the Northern krill body, which means it occurs by dietary (phytoplankton) accumulation through feeding.

The total MAA concentration in Northern krill eyes ranged from 0.72 to 2.06 mg g^−1^ d.w., with a mean of 1.87 mg g^−1^ d.w. (SD = ±0.88 mg g^−1^ d.w.), which was approximately two times higher than that in the remaining dissected body parts except for the gut (0.99 ± 0.41 mg g^−1^ d.w., ranging from 0.52 to 1.53 mg g^−1^ d.w.) (Figure 3A). In addition, there was variation in the percentage composition of the MAA types between the eyes and other body parts. In detail, four different MAAs (PA, PR, MG, and unidentified MAA; M−332) were isolated from the eyes of Northern krill, whereas all other body parts, except for the gut, only contained two different kinds of MAAs (SH and PR) (Figure 3B).

## 3. Discussion

### 3.1. Pigment-Based Phytoplankton Community Composition

Spring Chl-a concentrations showed considerable variability in Kongsfjorden; peak values of up to 4.9 μg L^−1^ were observed during the growing season [48]. Considering that the sampling period and our Chl-a data were not a spring bloom, the observed values were within the ranges that were previously published [48,49,50]. As shown in Figure 1, the mean Chl-a concentrations in transect A were relatively high in the middle layer, where the total accessory pigment concentrations were high. The probability was high that the shallowest Chl-a maxima would be found at transect T (except for T4) and its relationship with the adjacent land area was associated with high sediment load, which regulates light conditions [51,52,53].

Pigment analyses revealed that the phytoplankton community structure was markedly different between cross-sectional areas. In this study, Fuco and Allo were used to reflect diatoms and cryptophytes, respectively [41,44]; Per is the characteristic pigment of autotrophic dinoflagellates [41,54]. Phytoplankton assemblages in transect A were dominated by diatoms, as indicated by high Fuco concentrations. The pigment compositions of phytoplankton in transect T were dominated by Allo followed by Per, suggesting that the phytoplankton community consisted mostly of flagellated phytoplankton, particularly cryptophytes. This result mirrored the findings of Wright et al. [55]; they reported that a relatively high biomass of cryptophytes was observed under the sea ice and ice edge. Lizotte et al. [56] also noted that cryptophytes, seem to be associated with glacial melt water. However, Ha et al. [57] observed a low contribution of cryptophytes in phytoplankton composition during May in Kongsfjorden (at stations similar to ours) where *Phaeocystis* spp. dominated. The differences in phytoplankton communities may be due to the influence of oceanographic conditions. According to Hodal et al. [58], the phytoplankton community in Kongsfjorden is dominated by diatoms such as *Thalassiosira* spp. and *Chaetoceros* spp. in May, when Arctic water masses prevail. Increases in warm water intrusion into fjords have induced changes in the phytoplankton distribution and regime shifts from diatom dominance to a *Phaeocystis pouchetii* [58,59]. Based on these findings, in the present study, the presence of diatoms and absence of 19′-hexanolyoxyfucoxanthin (19′-Hex; a marker pigment for haptophytes, primarily *Phaeocystis* sp. [60]) in the central part of the fjord seem to be more strongly influenced by Arctic water masses flowing inward from the shelf, whereas the phytoplankton community in waters adjacent to land was more regionally affected by sea ice or glacier ice.

### 3.2. Properties of MAA in Phytoplankton

Five individual MAA compounds, SH, PA, PR, MG, and AS, are commonly found in phytoplankton [61]. Large variability in the MAA concentrations between the different water layers was observed (Table 1). SH and PA were the predominant MAAs in phytoplankton of the surface water, whereas MG was the main MAA in phytoplankton of middle-layer water. MG is the primary MAA produced through biosynthesis in the shikimate pathway and it acts as a precursor of SH ([62] and references therein). In addition, unlike many other MAAs, it absorbs within the UV-B range (280–320 nm) and presents high antioxidant activity, thus providing rapid protection against oxidative stress and photodynamic damage (reviewed in [28]). It was therefore concluded that at least some of the MAAs might play an important role in protecting phytoplankton from UV-R damage, not only by absorbing light energy and dissipating energy as heat, but also, probably more importantly, by scavenging photodynamically generated reactive oxygen species.

Overall, the measured total MAA concentration (sum of the individual MAAs) ranged from 0.3 to 4.3 μg L^−1^ with a mean of 1.3 μg L^−1^ in the surface layer compared with the middle-layer (0.2–1.2 μg L^−1^ with a mean of 0.5 μg L^−1^) (*t*-test, *p* < 0.05) (Table 1). In parallel with the total MAAs, the MAA to Chl-a ratio (MAA/Chl-a) on the surface of the fjord ranged between 0.7 and 10.7 μg (μg Chl-a)^−1^, with a mean of 3.3 μg (μg Chl-a)^−1^ (SD = ±2.6 μg (μg Chl-a)^−1^), which was approximately three times higher than that of the middle-layer water (mean ± SD = 0.9 ± 0.5 μg (μg Chl-a)^−1^) (Figure 2). However, the observed values are significantly lower compared with the values (mean of 36.3 μg (μg Chl-a)^−1^) from Ha et al. [50] in May in Kongsfjorden. Ha et al. [50] reported mean Chl-a concentrations (0.43 μg L^−1^) similar to our results, and the observed MAA/Chl-a difference could be due to the different phytoplankton communities studied between both studies. *Phaeocystis* sp. was dominant for Ha et al. [50], whereas diatoms and cryptophytes were dominant in our study. In addition, through field-based ^13^C labeling, they confirmed that *Phaeocystis* sp. produced higher MAA production rates than diatoms. This result mirrored the findings of [63,64]; they reported higher concentrations of UV-absorbing compounds in *Phaeocystis* than in diatoms.

However, we did not find an increase or decrease in the percentage of specific MAAs between two sections (transect A; dominated by diatoms and transect T adjacent land stations; dominated by cryptophytes), despite our results that show significant differences in the spatial distribution of phytoplankton. For example, the presence of diatoms can produce more SH and PR ([50,62] and references therein). This result suggests that these compounds primarily act as photoprotectants against harmful levels of UV-R, irrespective of their community structure, at least in the presence of diatoms or cryptophytes. Clearly, phytoplankton in the surface waters are most sensitive to UV-R because the qualitative and quantitative composition of MAAs in phytoplankton shows significant difference between phytoplankton in the surface and the middle depth. This is consistent with previous findings on various marine organisms from MAA-dependent UV-R doses [65,66,67,68]. As evidence, a broader range of MAA types in surface phytoplankton is synthesized as an important strategy to protect against a wide range of UV-R wavelengths. In contrast, the middle layer, as compared with surface water, has a relatively weak influence due to UV-R exposure conditions, and a precursor of the MAA compound (MG) accounted for more than 60% of the total MAA concentration.

### 3.3. Linkages between Phytoplankton Assemblages and Northern Krill

In our study, the higher relative Allo proportions in Northern krill might be a direct result of the fact that the food they consume is richer in cryptophytes. The Northern krill were caught under the sea ice where higher concentrations of Allo existed, and it accounted for the majority of the total contribution even in surface water areas around sea ice. Interestingly, as a minor pigment, 19′-Hex, as a marker pigment of *Phaeocystis* sp., was found in Northern krill. Hamm et al. [69] previously reported that *P. pouchetii* is the main food source for krill; however, pertinent to this study is the finding that cryptophytes are a major food source for Northern krill. This unexpected observation suggests that it may have been directly feeding on *Phaeosystis* sp. that lived in the ice. Another possible cause of the bioaccumulation of 19′-Hex could be through the marine food web. In response to prey diversity, krill species have evolved different strategies to cope by adaptive feeding behavior and functional responses. Previous studies have demonstrated that these organisms can actively exhibit species-specific selectivity and prey size preferences, switch from filter-feeding to ambush feeding, and modify their foraging efforts in response to the prey concentration ([70,71,72] and the references therein). For example, among North Atlantic krill communities, *Meganyctiphanes norvegica* and *T. raschii*, as omnivorous species, can switch from phytoplankton to zooplankton diets to quickly meet their energy requirements in high-latitude fjords [72,73]. However, in our study, the concentration of 19’-Hex was approximately one-hundredth of that of Allo and did not have much effect in inferring the main diet. Therefore, our results suggest that Northern krill may mainly consume the natural phytoplankton community (particularly cryptophytes) rather than obtaining Allo from other dietary sources; this may depend on the prey concentration.

### 3.4. Bioaccumulation and Distribution of MAAs in Northern Krill

From the results presented in Section 3.3, it is hypothesized that Northern krill feeds in nearby glacial and sea-ice areas. Therefore, in this section, based on the food availability of Northern krill, the concentration and distribution of MAAs will be explored in relation to the feeding activity by limiting the community structure (mainly cryptophytes) of the transect T surface region as the main feeding site.

The average total MAA concentration in whole krill varied twofold among the samples, ranging from a maximum of 5.8 mg g^−1^ d.w. to a minimum of 2.5 mg g^−1^ d.w. Riemer et al. [74] found similar values of total MAAs in the New Zealand krill *Nyctiphanes australis* (1.6 to 5.2 mg g^−1^ d.w.; mg g^−1^ was calculated assuming a unit molecular weight of SH). Karentz et al. [9] measured MAA concentrations in Antarctic krill *Euphausia superba* as 1.4 mg g^−1^ d.w., which were lower than those observed in krill in our study. The total MAA levels observed in the present study were within the range found in krill and/or other zooplankton [37], which may reflect the bioaccumulation of MAAs among our single species, intraspecific variability, and other environmental conditions.

Many studies have reported remarkable differences in the MAA concentrations and compositions of algae and animals in natural or experimental environments, which are affected by simulating abiotic factors (i.e., light, osmotic, and temperature) and biotic interactions ([75,76] and references therein). Newman et al. [33] show that the concentrations and proportions of MAAs in krill are dependent on the MAA content of phytoplankton and the algae’s physiological response to visible and UV-R exposure. Exposure to UV-B can change the nutritious quality of phytoplankton in terms of food for copepods, the main diatom grazers [77]. This is in line with observations indicating the dietary accumulation of MAAs and their consequent translocation and transformation in various organisms (i.e., zooplankton, metazoans, and fish) have been studied [74,78,79]. For example, Riemer et al. [74] showed that *N. australis* had similar concentrations of MAAs throughout the body, ranging from 1.6 to 2.0 mg g^−1^ d.w., with the exception of the carapace. Higher MAA concentrations were found in the eggs than in other body areas for the copepod *Cyclops abyssorum tatricus* [79].

We compared the levels of the individual MAAs between the eyes and the other body parts. This is an important observation on how MAAs are distributed throughout the body as a potential defense mechanism against UV-R. Indeed, the MAAs of Northern krill populations differed not only in total concentration, but also in the composition of individual compounds in their various body parts. Overall, PR was found in the highest concentrations in the body and eyes. Interestingly, SH and PA were the principal MAAs in the surface phytoplankton of transect T, although PA was not found in the body part (without the gut and eyes) of Northern krill, which indicates the transport specificity fraction of MAAs. Carroll and Shick [80] found that in sea urchins, high concentrations of specific MAAs present in certain tissues may have been a result of the removal of MAAs by more than 99% during passage through gut- and sex-specific accumulation related to spawning (particularly the ovaries). They also demonstrated that the presence of MG, despite its absence from the experimental meal, may reflect its production from SH or other algal MAAs by marine bacteria. One study provided information on some MAAs (such as SH) that are more efficiently translocated than others [16]. Another possibility is that a time lag between the synthesis of MAAs and the subsequent accumulation of these compounds in copepods [81] could potentially explain the difference between the composition of phytoplankton and Northern krill.

Most importantly, krill eyes have considerably more diverse MAAs and higher concentrations than other body parts, and these compounds help prevent UV damage. These results indicate that the eyes are the body part most sensitive to light, and relatively small amounts of MAAs appear to be present in the body, because body parts with shells in Northern krill mainly block harmful UV-R. Indeed, Riemer et al. [74] observed lower carapace MAA concentrations (0.7 mg g^−1^ d.w.) than in other body parts in *N. australis*. The exoskeletons of crustacean shells mainly comprise chitin, which has a protective effect against UV-R and acidification [82]. Thus, differing compounds of MAAs in the body in natural populations of Northern krill are probably an important strategy to minimize UV damage and their physiological demands.

In summary, in this study, the dietary source of Northern krill seems to be closely associated with close proximity to the glacier front rather than with water depth in our study area of Kongsfjorden, Svalbard. Several lines of evidence indicate that cryptophytes are the most primitive food source for Northern krill and contribute to MAAs, supporting our hypothesis. The relative contribution of the different compounds to the total MAA pool in Northern krill varied among the body parts. The eyes of Northern krill contain a high concentration and various types of MAAs (SH, PA, PR, and M-332), whereas SH and PR and lower concentrations of MAAs were only observed in the body (without the gut and eyes) of krill. However, MAA compositions in Northern krill are not dependent on the MAA proportion of phytoplankton on the transect T surface, which may be due to differences in the MAA-specific kinetics or the translocation of MAA concentrations or other biological processes such antioxidant defense mechanism, osmotic regulation, and many other cellular functions. Therefore, additional data such as stable isotopes or fatty acids to identify food sources for Northern krill and other environmental factors inducing MAA synthesis will be needed for further investigation, to enable a more in-depth discussion.

## 4. Materials and Methods

### 4.1. Study Area and Sampling

Kongsfjorden is a sub-Arctic inlet on the west coast of Spitsbergen in Svalbard, which is 26 km long and 6–14 km wide. It is a key open fjord and has a large impact on biodiversity and animal populations with physical and biological variations between the Arctic and Atlantic waters ([49] and references therein). This research was conducted onboard Teisten at nine stations (A1–A4, T1–T5) in Kongsfjorden from 22 May to 31 May 2011 (Figure 4). Sampling was conducted along two transects: transect A across Kongsfjorden from the mouth of the fjord toward the shelf (five stations: A1, A2, A3, and A4) and transect T between adjacent land stations with land-fast ice edges (five stations: T1, T2, T3, T4, and T5) (Figure 4). The mouth of the fjord is more influenced by oceanic conditions than the innermost part; the innermost part of the fjord is strongly affected by melted freshwater, sediment input, and drifting ice from glaciers.

Water samples were collected from the surface and middle-layer (depth of 30 m) waters for the MAAs and the pigments of phytoplankton using a Niskin sampler. At T1, seawater samples were only obtained from the surface layer due to the shallow water. Triplicate 1 L water samples were filtered through a preignited Whatman GF/F filter (25 mm) and then immediately stored at −80 °C for later analysis. Northern krill (*Thysanoessa* sp.) samples were collected from the sea-ice edge (Alfred-Wegener-Institute, Germany), flash-frozen in liquid nitrogen, and stored at −80 °C for the later extraction of MAAs and pigments.

### 4.2. Pigment and MAA Analyses

For the pigment extraction of phytoplankton and gut contents of Northern krill, samples were transferred into Teflon tubes with 3 mL of 100% acetone, and ultrasonic extraction (30 s, 50 W) was applied; then, they were stored in a cool, dark place for 24 h. An internal standard (50 μL of apo-8-carotennoate) was used to compensate for the losses during extraction. Extracted pigments in the solvent were filtered (0.2 μm hydrophobic syringe filter), and then a 1 mL aliquot of the sample was taken and 300 μL of deionized water (DIW) was added. The chromatographic separation and quantification of pigments were conducted on an Agilent 1200 series HPLC (Santa Clara, CA, USA) using a Waters symmetry C8 column (150 × 4.6 mm, 3.5 μm, Milford, CT, USA) [83] with mobile phase A (methanol: 50%, acetonitrile: 25%, aqueous pyridine solution: 25%) and mobile phase B (methanol: 20%, acetonitrile: 60%, acetone: 20%). Pigments were identified by the retention times and absorption spectra of the corresponding standards (Chlorophyll a, b (Sigma-Aldrich, Saint Louis, MO, USA), c2, c3, Peridinin, 19′-butanoyloxyfucoxanthin, Fucoxanthin, Neoxanthin, Prasinoxanthin, Vioxanthin, Diadinoxanthin, Alloxanthin, Zeaxanthin, β-Carotenoid, and 19′-hexanolyoxyfucoxanthin (DHI Water and Environment, Hørsholm, Denmark)) [84]. These standard pigments were calculated with the method of Park and Park [85], using extinction coefficients [41]. In order to distinguish the major phytoplankton taxa, four marker pigments were used in the present study: Fucoxanthin for diatoms, Alloxanthin for cryptophytes, Peridinin for flagellates, and 19′-hexanolyoxyfucoxanthin for *Phaeocystis* sp.

The bodies of the Northern krill were separated into two parts for the MAA analysis: the eyes and remaining body parts, except for the gut. Gut contents from the Northern krill were removed, including the entire digestive gland. For MAA extraction, 3 mL of 100% methanol was added to phytoplankton and Northern krill, which were sonicated using an ultrasonicator (30 s, 50 W; ULH-700s, Ulsso Hi-tech, Cheongju, Korea) and then left overnight at 4 °C. Extraction and filtration were performed using a 0.2 μm syringe filter (PTFE Hydrophobic, Advantec Toyo Roshi Kaisha, Tokyo, Japan). Subsequently, 2 mL of extraction solution was transferred to a microtube and then dried using a centrifugal evaporator (EYELA, CVE-200D, Tokyo, Japan). The completely dried sample was soaked in 500 μL of DIW, and then 100 μL of chloroform (for the removal of pigments and lipids) was added. The supernatant (400 μL) was collected by centrifugation at 10,000 rpm for 10 min. MAA concentrations were determined using high-performance liquid chromatography (HPLC). The detector used an Agilent DAD (G1315D, Santa Clara, CA, USA), and the wavelength was 313 nm (250–750 nm scan). The MAAs were separated using a Waters 120DS-AP (5 μm) 150 mm × 4.6 column. The mobile phase of HPLC used 0.1% acetic acid in 100% DIW at a constant flow rate of 0.8 mL min^−1^. The standard material used for MAA analysis was provided by Professor Häder of Freiderich Alexender University in Germany.

## Figures and Tables

**Figure 1 marinedrugs-20-00238-f001:**
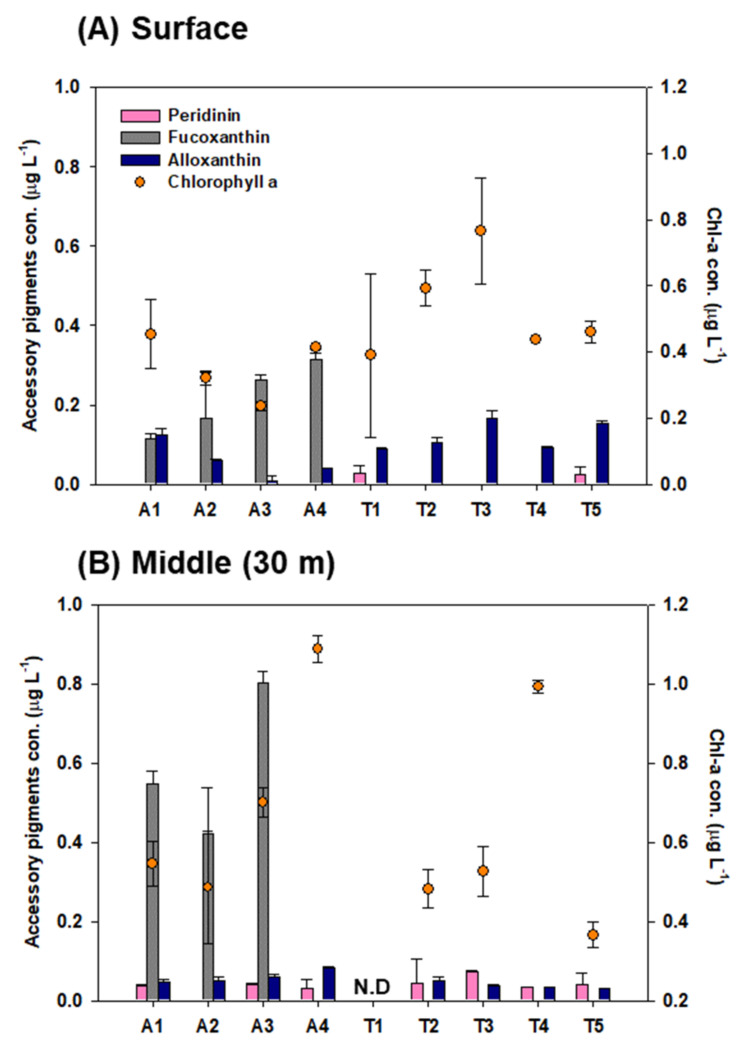
Concentration of chlorophyll a and accessory pigments of phytoplankton at (**A**) surface and (**B**) middle-layer (30 m) waters in the Kongsfjorden. The bar charts show mean concentrations ± standard deviation of pigments. N.D indicates no data.

**Figure 2 marinedrugs-20-00238-f002:**
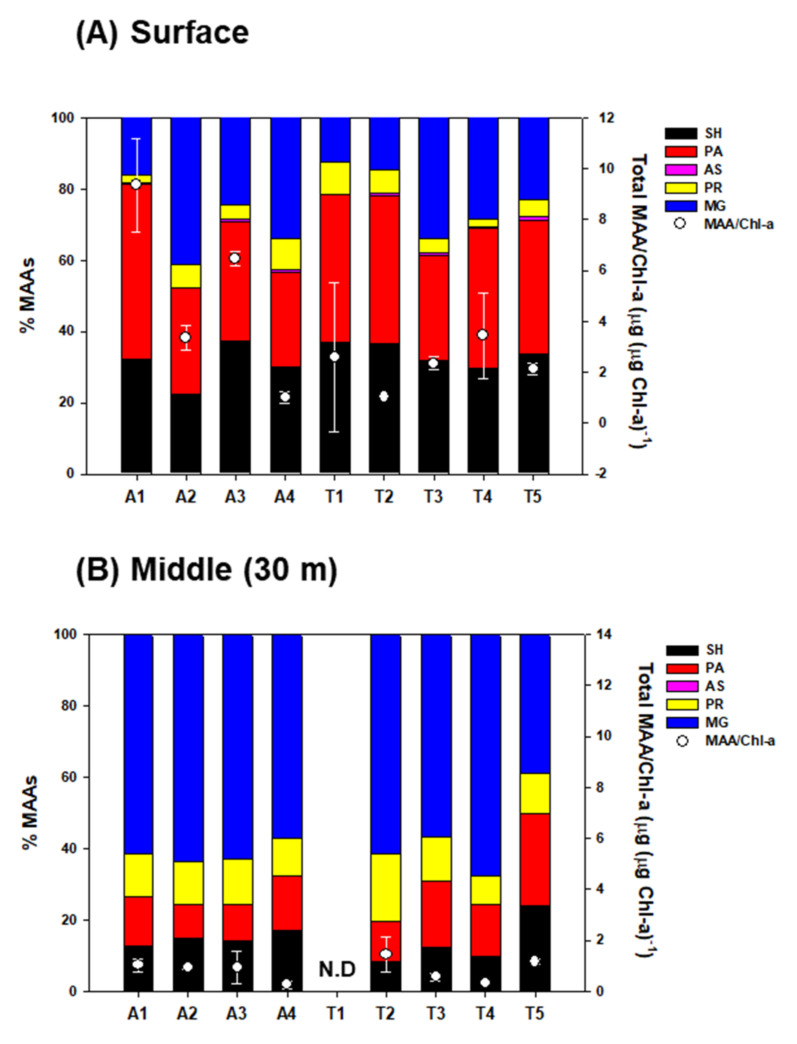
Relative contribution (%) of the different compounds to the total mycosporine-like amino acid (MAA) contents at two layers ((**A**) surface and (**B**) middle-layer water) in phytoplankton. Dots and error-bars represent the mean MAA/Chl-a and standard deviation (*n* = 3), respectively. SH; Shinorine, PA; Palythine, AS; Asterina-330, PR; Porphyra-334, and MG; Mycosporine-glycin. N.D indicates no data.

**Figure 3 marinedrugs-20-00238-f003:**
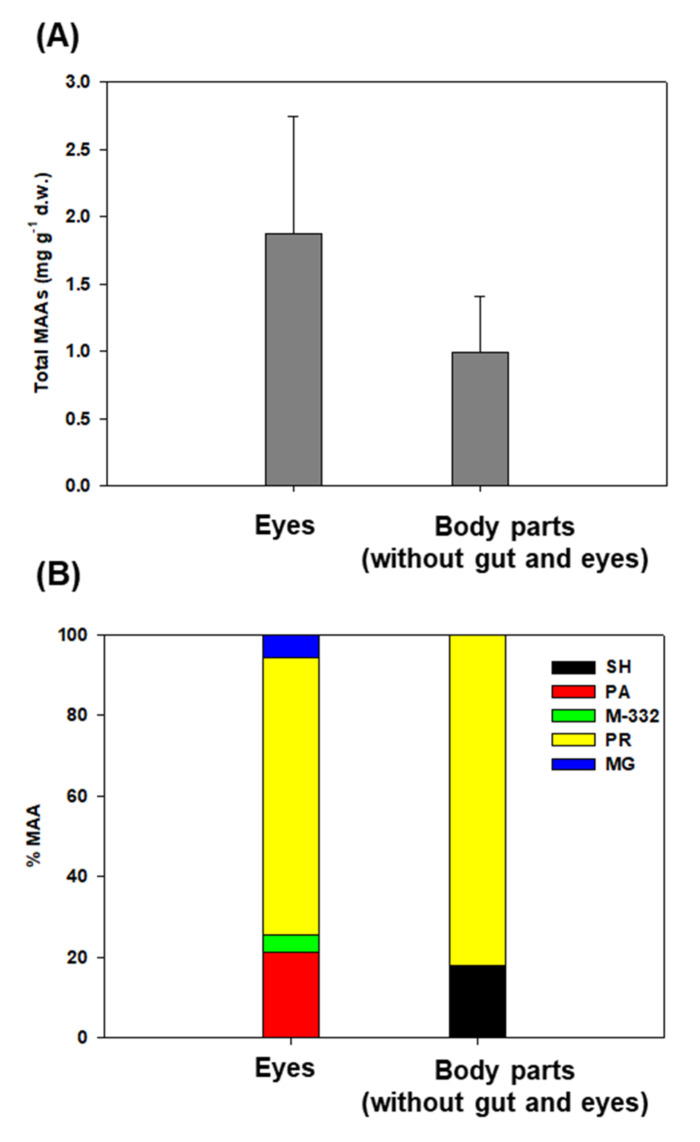
Northern krill (*Thysanoessa* sp.) from Kongsfjorden, collected in 2011. (**A**) Body distribution of total concentration MAAs and (**B**) composition of individual MAAs of Northern krill (MG; Mycosporine-glycin, PR; Porphyra-334, M-332; unknown compound λmax: 332 nm, PA; Palythine, and SH; Shinorine).

**Figure 4 marinedrugs-20-00238-f004:**
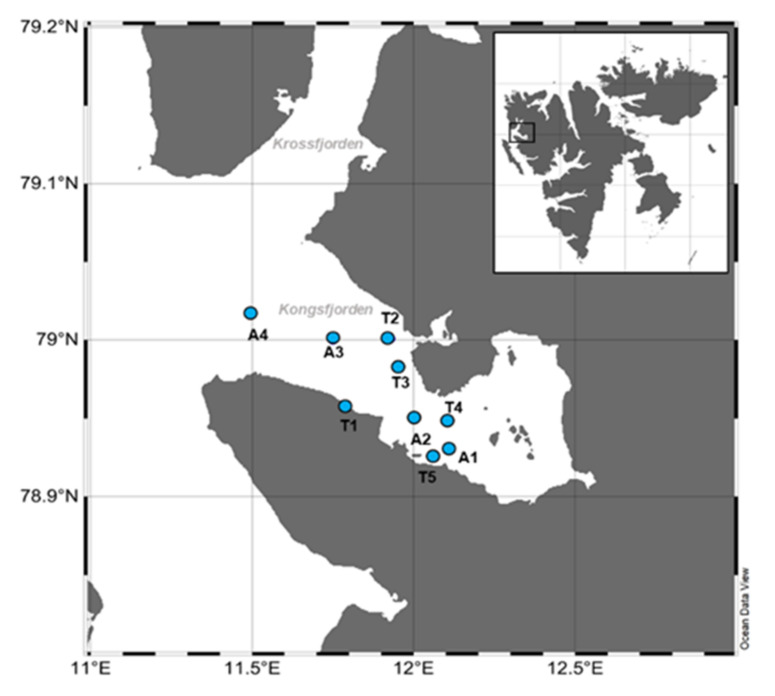
Location of sampling sites in Kongsfjorden.

**Table 1 marinedrugs-20-00238-t001:** Mean (± standard deviation, *n* = 3) concentrations of individual MAA compounds (μg L^−1^) in: (A) surface and (B) middle-layer waters in Kongsfjorden. At T4 (30 m depth), MAA was measured once due to sample loss.

Station	MAAs (μg L^−1^)				
	SH	PA	AS	PR	MG
(A) Surface					
A1	1.349 (±0.033)	2.029 (±0.059)	0.023 (±0.001)	0.094 (±0.001)	0.662 (±0.172)
A2	0.241 (±0.018)	0.318 (±0.018)		0.069 (±0.011)	0.448 (±0.098)
A3	0.565 (±0.033)	0.506 (±0.031)	0.014 (±0.001)	0.056 (±0.002)	0.371 (±0.014)
A4	0.128 (±0.030)	0.113 (±0.032)	0.003 (±0.000)	0.036 (±0.005)	0.145 (±0.048)
T1	0.192 (±0.029)	0.216 (±0.025)		0.048 (±0.012)	0.074 (±0.069)
T2	0.226 (±0.031)	0.254 (±0.034)	0.006 (±0.001)	0.039 (±0.008)	0.090 (±0.031)
T3	0.556 (±0.175)	0.506 (±0.101)	0.014 (±0.002)	0.066 (±0.017)	0.580 (±0.109)
T4	0.406 (±0.112)	0.527 (±0.120)	0.014 (±0.002)	0.028 (±0.007)	0.533 (±0.485)
T5	0.331 (±0.063)	0.366 (±0.047)	0.008 (±0.001)	0.047 (±0.011)	0.226 (±0.048)
(B) Middle (30 m depth)					
A1	0.069 (±0.010)	0.075 (±0.009)		0.065 (±0.013)	0.354 (±0.136)
A2	0.064 (±0.006)	0.041 (±0.005)		0.052 (±0.017)	0.284 (±0.085)
A3	0.096 (±0.077)	0.067 (±0.048)		0.077 (±0.042)	0.405 (±0.274)
A4	0.041 (±0.004)	0.036 (±0.007)		0.028 (±0.009)	0.180 (±0.145)
T2	0.054 (±0.018)	0.070 (±0.017)		0.147 (±0.125)	0.441 (±0.231)
T3	0.036 (±0.011)	0.051 (±0.009)		0.039 (±0.018)	0.178 (±0.098)
T4	0.016	0.024		0.014	0.113
T5	0.102 (±0.003)	0.109 (±0.022)		0.047 (±0.008)	0.167 (±0.046)

**Table 2 marinedrugs-20-00238-t002:** Chlorophyll a and accessory pigment concentrations (μg g^−1^ d.w.) in Northern krill.

Diagnostic Pigment	Abbreviation	Concentration (μg g^−1^ d.w.)	Phytoplankton Group(s)
Fucoxanthin	Fuco	0.016 ± 0.002	Diatoms
19′-hexanolyoxyfucoxanthin	19′-Hex	0.029 ± 0.009	*Phaeocystis* sp.
Alloxanthin	Allo	2.791 ± 0.333	Cryptophytes
Chlorophyll a	Chl-a	8.286 ± 1.127	

## Data Availability

The original contributions presented in the study are included in the article, further inquiries can be directed to the corresponding author.

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
