# Peer review of "The Interplay of Mycosporine-like Amino Acids between Phytoplankton Groups and Northern Krill (Thysanoessa sp.) in a High-Latitude Fjord (Kongsfjorden, Svalbard)"

_marinedrugs, 2022, doi:10.3390/md20040238_

Round 1

Reviewer 1 Report

General comment My main concern is that the paper describes results obtained 10 years ago. The samples were collected in May 2011 and analysed also at that time (?). Why were not published the results earlier? Do the results have any relevance today? The references are quite old, only few recently published papers are cited. The paper doesn’t have real novelty. It is well known that aquatic animals can accumulate MAAs. This paper confirms this establishment by using northern krill and highlights only minor differences compared to the published results of other researchers (Riemer at al. 2007).

Abstract The abstract is informative, underlines some results and summarizes the main establishment.

Methods The pigment-based analysis could provide only suggestions about the phytoplankton community composition. Nowadays, precise metagenomic analysis are available for detection of phytoplankton composition. I miss the measurements of the UV radiations, which is the main factor influencing the production of MAAs.

Results and discussion This chapter is well structured and clearly described. The results supported the main establishment: the northern krill accumulates MAAs from the diet, but it could not be specified.

In spite of the above mentioned comments, the paper is well structured, the results are clearly described.

Author Response

General comment My main concern is that the paper describes results obtained 10 years ago. The samples were collected in May 2011 and analysed also at that time (?). Why were not published the results earlier? Do the results have any relevance today? The references are quite old, only few recently published papers are cited. The paper doesn’t have real novelty. It is well known that aquatic animals can accumulate MAAs. This paper confirms this establishment by using northern krill and highlights only minor differences compared to the published results of other researchers (Riemer at al. 2007).

Abstract The abstract is informative, underlines some results and summarizes the main establishment.

Methods The pigment-based analysis could provide only suggestions about the phytoplankton community composition. Nowadays, precise metagenomic analysis are available for detection of phytoplankton composition. I miss the measurements of the UV radiations, which is the main factor influencing the production of MAAs.

Results and discussion This chapter is well structured and clearly described. The results supported the main establishment: the northern krill accumulates MAAs from the diet, but it could not be specified.

In spite of the above mentioned comments, the paper is well structured, the results are clearly described.

--> Thank you for your kind comments on the manuscript. Many studies have been conducted on krill in Antarctica, and research on body composition has also been studied. However, there are relatively few studies on Arctic krill, so it is considered to be an important data for ecological changes due to climate change. Analysis was completed simultaneously with sampling in 2011. As you mentioned, better results could have been obtained if metagenome analysis and measurements of the UV radiations were performed, but more analysis parameters could not be included in conducting the research within a limited time. We will proceed with sample collection and analysis taking this into consideration when conducting future research.

Reviewer 2 Report

This manuscript has some interesting data and an equally interesting topic, but the writing needs a serious upgrade. It falls apart once you get to the third paragraph of the Results and Discussion section, and is problematic through the rest of that section, becoming increasingly hard to follow.  I strongly suggest the authors make separate Results and Discussion sections; this paper is too long to combine them. I also suggest that they engage a native speaker or someone that writes well to help sort out their composition problems.

Some specific problems/questions:

1. What does the filtration step in the processing of the water samples do to the data? Why is it done?

2. Is there any dissolved MAAs in the water, or they all presumed to be in the phytoplankton cells? Also, please be clear when referring to water layer versus phytoplankton.

3. I don't understand the reference to Table 2 about "the distinct order of appearance ..." (lines 285 and 286)? Order implies time or phases, and there is no such data in the paper.

4. I don't understand the sentence that begins on line 205 with "Considering...". How is this a reasonable result? 

Specific Suggestions for writing:

  1. Please avoid gratuitous (unnecessary) words and phrases, and minimize redundant words and phrases, such as "in our study".
  2. Please check verb tenses, is versus was, etc.
  3. Use a grammar checker such as in MS Word.  For example, line 178 needs a "which" or "that" between "results" and "show". Also, in that same sentence, "community" is gratuitous. 
  4. Be careful of what you put in parentheses. Many times the numbers should be just stated in the sentence and not put in parentheses (cf. the paragraph after Table 2).  

Author Response

#Reviewer 2

This manuscript has some interesting data and an equally interesting topic, but the writing needs a serious upgrade. It falls apart once you get to the third paragraph of the Results and Discussion section, and is problematic through the rest of that section, becoming increasingly hard to follow.  I strongly suggest the authors make separate Results and Discussion sections; this paper is too long to combine them. I also suggest that they engage a native speaker or someone that writes well to help sort out their composition problems.

--> Thank you for the detail comment on the manuscript. MDPI English editing service was done on the entire manuscript. As you suggested, we separated “Results and Discussion” section and revised thoroughly in text.

Some specific problems/questions:

  1. What does the filtration step in the processing of the water samples do to the data? Why is it done?
  2. Is there any dissolved MAAs in the water, or they all presumed to be in the phytoplankton cells? Also, please be clear when referring to water layer versus phytoplankton.

Responses 1 and 2: For analysis of phytoplankton in seawater, particles that were not filtered through a GF/F (ca. nominal pore diameter = 0.7 μm) filter were considered phytoplankton. Therefore, dissolved forms are not covered in this study. The surface (0m) or middle (30m) water layer means phytoplankton living in each water depth.

  1. I don't understand the reference to Table 2 about "the distinct order of appearance ..." (lines 285 and 286)? Order implies time or phases, and there is no such data in the paper.

--> To avoid any confusion, we deleted the sentences.

  1. I don't understand the sentence that begins on line 205 with "Considering...". How is this a reasonable result? 

Specific Suggestions for writing:

  1. Please avoid gratuitous (unnecessary) words and phrases, and minimize redundant words and phrases, such as "in our study".
  2. Please check verb tenses, is versus was, etc.
  3. Use a grammar checker such as in MS Word.  For example, line 178 needs a "which" or "that" between "results" and "show". Also, in that same sentence, "community" is gratuitous. 
  4. Be careful of what you put in parentheses. Many times the numbers should be just stated in the sentence and not put in parentheses (cf. the paragraph after Table 2).  

--> We have corrected them.
